# The Effect of Acyl Chain Position on the 2D Monolayer Formation of Monoacyl-*sn*-Glycerol at the Air/Water Interface: Quantum Chemical Modeling

Elena S. Kartashynska [1,2]

1   L.M. Litvinenko Institute of Physical Organic and Coal Chemistry, 70 R. Luxemburg Str.,
    283050 Donetsk, Russia; elenafomina-ne@yandex.ru; Tel.: +7-949-388-56-57
2   Department of General, Physical and Organic Chemistry, Donetsk National Technical University,
    58 Artema Str., 283000 Donetsk, Russia

**Abstract:** This paper deals with the results of quantum chemical modeling of the monoacyl-*sn*-glycerol 2D cluster formation at the air/water interface using a semi-empirical PM3 method. The impact of the 2 or 3 positions of the acyl substituent on the thermodynamics of the monolayer formation is assessed for surfactants with an acyl substituent $C_nH_{2n+1}COO$ chain length of $n = 6$–17 carbon atoms. The calculation shows a significant change in the spontaneous clusterization threshold for isomeric compounds, which differs only in the position of the acyl substituent with respect to the glycerol backbone. This change is almost equal to substituent shortening by approximately two methylene fragments. At the same time, the geometric parameters of the unit cell for resulting monolayers are not affected so drastically. The 2D films in question possess an oblique or orthorhombic unit cell with parameters for 2 and 3-monoacyl-*sn*-glycerol monolayers, as follows: $a = 4.91$ Å and 4.82 Å and $b = 5.00$ Å and 4.92 Å, with hydrocarbon chains tilted at $t = 23.0°$ and $23.5°$. The calculated results are in accordance with existing experimental data obtained using grazing incidence X-ray diffraction measurements and the π-A isotherm technique.

**Keywords:** surfactant; 2D film; clusterization thermodynamics; semi-empirical PM3 method

## 1. Introduction

Monoacyl-substituted glycerols are numerous objects in the field of studying the structure and properties of monomolecular films, since these compounds are used as emulsifiers and included in pharmaceuticals and food products [1,2]. It should be taken into account that the processes occurring at the interface are complex and depend on many factors: both the external environment (temperature, pH, and the presence of different natural additives in the aqueous phase) and the structural peculiarities of the particular class of surfactants. In particular, the position of the substituent relative to the hydrophilic "head" of the surfactant or the functional group relative to the hydrophobic "tail" affects the structural characteristics of the resulting monolayers [3]. For example, in the case of studying monolayers of hydroxylated carboxylic acids at the water surface, the authors of papers [4–9] recorded a significant difference in the structural and thermodynamic parameters of the resulting films. With regard to monoacyl-substituted glycerols, the experiments performed and reported in [10] using the Langmuir trough technique reveal that the plateau regions of the π-A isotherms for two compounds, which differ only in the relative position of the acyl substituent along the glycerol skeleton, are shifted profoundly within the temperature scale. A similar pattern was recorded for monolayers of 2-methyl-branched alcohols [11]. In paper [12], vapor–liquid nucleation and the freezing of nanodroplets of short-chained alcohols with hydroxylic groups at the 1 and 3 positions were investigated. The authors revealed that 1-isomers perturbate freezing less than 3-isomers do, which indicates their less dense packing at the water surface. In the study of [13], the

impact of the hydroxylic group position on the dynamic adsorption of aliphatic alcohols is considered.

The most experimentally studied representatives of the class of monoacyl-substituted glycerols are palmitoyl and stearoyl homologs of both enantiomerically pure compounds and racemic mixtures [14–18]. There are significantly fewer theoretical studies modeling the film formation of monoacyl-substituted glycerols. These include Reference [19], which is devoted to the aggregation of monooctyl- and monooctadecylglycerol at the air/water interface. However, no systematic theoretical studies have been carried out for the homolog series of these surfactants with different substituent positions. Earlier, in [20], calculations of the thermodynamic and structural parameters of film formation for 3-monoacyl-*sn*-glycerols were carried out in the framework of a quantum chemical approach using the semi-empirical PM3 method. The present study aims to assess the effect caused by changing the acyl ($C_nH_{2n+1}COO$ with $n$ = 6–17) substituent position with respect to the glycerol skeleton from the third to the second position on the thermodynamics and geometry of monoacyl-*sn*-glycerol film formation at the air/water interface.

## 2. Calculation Methodology

Calculations of the targeted thermodynamic and geometrical parameters for mono-substituted glycerols were carried out using supermolecule approximation. Initially, this approach was applied for an account of solvation, where a system consisting of a solvable molecule and a certain limited number of solvent molecules is calculated using the quantum chemical method as a single molecule [21–24]. Later, this approach was also applicable to other molecular systems [25].

Within the framework of the exploited model, quantum chemical calculations are performed in a vacuum (assuming that the results for a vacuum are close to the results in air, since their dielectric permittivities are close), and the influence of the aqueous phase is taken into account through its stretching and orienting action. That is, when constructing the initial supermolecule, all the surfactant molecules of the condensed monolayer are assumed to be in the most elongated all-trans conformation. Since surfactant molecules are immersed in water within their "heads" and possibly several adjacent methylene units, they are arranged in a "mattress"-like structure floating on the water. In such a case, the larger part of the surfactant is present in the air phase, not in the water. Attempts to use the COSMO model, which takes into account the presence of a solvent in the system, in the example of aliphatic alcohols [26], have shown that the dimerization enthalpy calculated within this model and the dimerization enthalpy calculated in a vacuum differ slightly. In this regard, the consideration of the interface is indirect, which manifests itself in its orienting and stretching action. In a later work by Vysotsky et al. [27], the water phase was explicitly accounted for in the case of the aliphatic alcohol cluster formation of the hexagonal structure. This new scheme provides a slightly better agreement between the calculated and experimental values compared to the older one. But, significant differences in the threshold chain length for alcohol clusterization were not observed. Its value was 12 carbon atoms in the chain for both schemes. It should be noted that the proposed model, without explicitly accounting for the water phase, is suitable for any interface, provided the surfactants are in a nonionized form. In the present work, traditional Langmuir monolayers at the air/water interface for nonionic surfactants are considered.

The possible conformations of a hydrophobic surfactant chain should also be discussed. The surfactant chains can certainly possess different conformations beyond only the most extended ones. But, the present study, as well as other papers devoted to the 2D monolayer formation of surfactants in the framework of the proposed quantum chemical approach, is focused on the possibility of the formation of a condensed phase for corresponding monolayers. In this case, the surfactant hydrophobic chains are mainly in the maximum extended "linear" conformation. This fact was proven in numerous studies (for example, [28,29]) and can also be concluded from the monolayer thickness, which corresponds to the surfactant chain length accounting for the possible molecular tilt angle

with respect to the normal to the interface depending on the hydrophilic "head" dimensions [30–33]. Also, it is easy to confirm using grazing incidence X-ray diffraction data for surfactant monolayers. In the experimental study of [18], there are data for molecular in-plane area $A_{xy}$, the cross-sectional area of alkyl chain $A_0$, and molecular tilt angle *t* for 1-monostearoyl-*rac*-glycerol. There is a dependence $A_0 = A_{xy} \cdot \cos(t)$, while values of $A_0$ are within the scope of 19.1–19.6 Å$^2$. In paper [34], the $A$ is the area per molecule in the frozen surface layer for alkanes with a chain length of 18–44 carbon atoms. It varied within the range of 18.71–19.81 Å$^2$ corresponding to the linear conformation of the chain (see Figure 14 therein [34]).

The structure optimization of the monomers and surfactant clusters is carried out using the quantum chemical package Mopac2000 [35] within the semi-empirical PM3 method. Optimization is controlled by the value of the final gradient norm parameter equal to $10^{-5}$. The frequencies of harmonic oscillations are further calculated, enabling a further estimate of thermodynamic parameters. The absence of imaginary values of analytical harmonic vibrational frequencies is the criterion for considering found stationary points as the minima. Clusterization enthalpy, entropy, and Gibbs energy for associates are calculated using the following equations: $\Delta H_{T,m}^{Cl} = \Delta H_{T,m}^{0} - m \cdot \Delta H_{T,mon}^{0}$, $\Delta S_{T,m}^{Cl} = S_{T,m}^{0} - m \cdot S_{T,mon}^{0}$, and $\Delta G_{T,m}^{Cl} = \Delta H_{T,m}^{Cl} - T \cdot \Delta S_{T,m}^{Cl}$, where $\Delta H_{T,m}^{0}$ and $S_{T,m}^{0}$ are enthalpy and absolute entropy of the formation of corresponding clusters at the temperature T; $\Delta H_{T,mon}^{0}$ and $S_{T,mon}^{0}$ are the corresponding parameters of formation for the monomers at the same temperature T; and m is the number of monomers in the cluster.

Obviously, the PM3 method has certain limitations [36]. It is parameterized within formation heats using data for a number of small isolated molecules. Its applicability for cluster calculations should be speculated about and can be approved, provided the results obtained by means of PM3 agree with the experimental data or those using more accurate methods. These data exist in the literature, although they concern mostly dimers of the first representatives of the alkane homologous series. For example, in Ref. [37], devoted to ab initio calculations for methane and neopentane dimers, Metzger et al. have shown that on the potential energy surface of methane and neopentane dimers, there are minima corresponding to the occurrence of CH⋯HC interactions. But, the depth of these minima is about 2 times lower than for the minima obtained using the PM3 method. In the same paper, the authors noted that three studied semi-empirical methods, MNDO, AM1, and PM3, differently consider intermolecular interactions. It was theoretically shown that only an adequate account for the correlation energy with the correct Hamiltonian makes it possible to describe intermolecular interactions satisfactorily. There is no explicit or implicit account of dispersion interactions in the MNDO method. In the AM1 method, the nuclear repulsion function was modified, which resulted in a better description of the interaction energy of alkane dimers than in the MNDO. As for the PM3 method, all its parameters differ significantly from those used in the MNDO and AM1 methods, since they have undergone full optimization. The results obtained by Metzger for methane and neopentane dimers using the PM3 method are not the result of modification of the nuclear repulsion function. In this regard, it is difficult to recommend one or another semi-empirical method for describing dispersion interactions. As for the later modifications of the PM3, such as PM6 and PM7 methods, they give almost identical results according to the values of the thermodynamic parameters of clusterization of the considered systems. However, they underestimate the length of CH⋯HC bonds even more than PM3, which opposes the calculation of the geometric parameters of the unit cells of surfactant monolayers. The criterion for the applicability of a particular method is in agreement with the experiment, which is the case for the PM3 method. It adequately describes the threshold lengths of surfactant chains, the temperature effect of clusterization, the geometric parameters of monolayer unit cells for more than 10 classes of surfactants, as well as changes in the surface acidity or basicity of saturated carboxylic acids and amines depending on their chain length [38].

### 3. Results and Discussion

#### 3.1. Conformational Analysis of Monomers

The investigation starts with the conformational analysis of monomers for targeted compounds. An analysis of possible conformations of a glycerol molecule was carried out in a previous paper [20]. There are two possible conformers of glycerol molecules. Later, these glycerol conformers were used to obtain 2 or 3-monoacyl-*sn*-glycerol monomers by replacing its second or third hydroxylic group with a $C_nH_{2n+1}COO$ substituent. The structure optimization of the glycerol molecule includes varying the torsion angles for all three hydroxylic groups: at the first $\angle C_1C_2O_2H_2$, the second $\angle C_2C_3O_3H_3$, and the third $\angle C_2C_1O_1H_1$ carbon atoms of the glycerol skeleton, respectively (Figure 1a), within the step of 10°. As a result, two stable conformations of glycerol are found, both of which contain intramolecular hydrogen bonds depicted with blue dashed lines in Figure 1a. In terms of the heat of formation, the structure of the glycerol conformer 2 (with the heat of formation −605.5 kJ/mol) better corresponds to the experimental value (−607.5 kJ/mol, obtained using glycerol heat of formation in liquid form at −668.5 kJ/mol and its heat of vaporization at 61.0 kJ/mol) [39]. For this structure of glycerol monomer, $\angle C_1C_2O_2H_2 = -40°$, $\angle C_2C_3O_3H_3 = -60°$, and $\angle C_2C_1O_1H_1 = 67°$. Therefore, this glycerol conformer 2 is further used to construct monomers of 2 and 3-monoacyl-*sn*-glycerols.

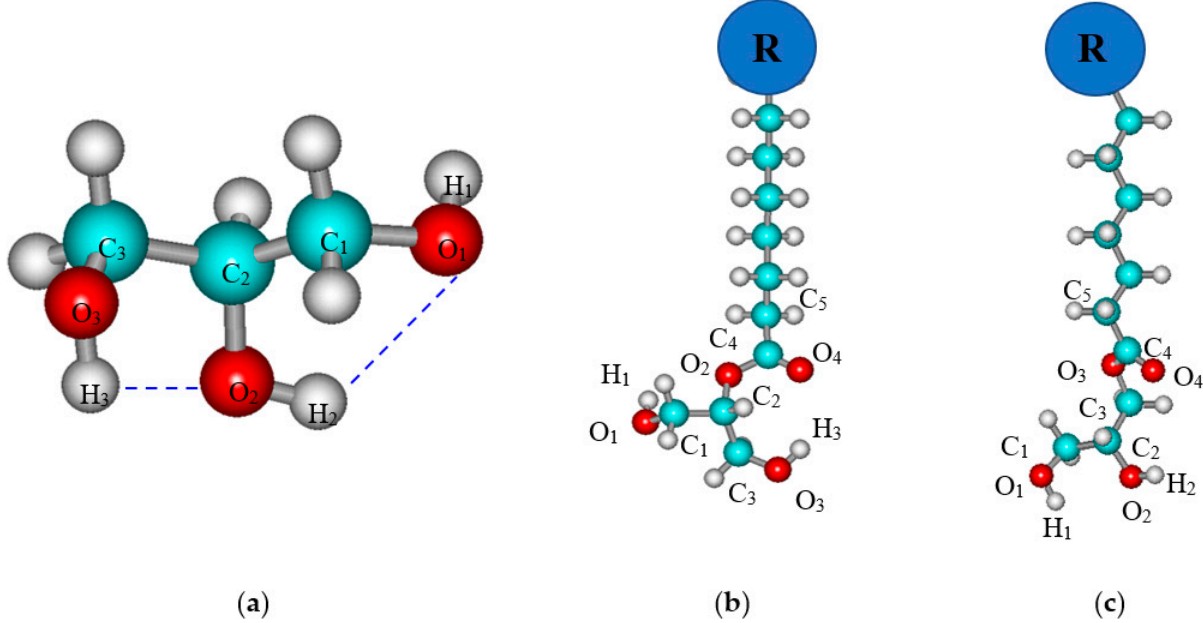

(**a**)          (**b**)          (**c**)

**Figure 1.** Structures of the most advantageous conformer 2 of glycerol (**a**), 2 and 3-isomers of monoacyl-*sn*-glycerol (**b**) and (**c**) correspondingly (R is the alkyl chain).

Figure 1b,c show the structures of monomers 2 and 3-monoacyl-*sn*-glycerols, which are subjected to further conformational analysis. The structure of the 2-monoacyl-*sn*-glycerol monomer is considered the first. The values of the torsion angles of the 1 and 3 positions of hydroxylic groups relative to the glycerol skeleton are first analyzed again. As can be seen at the surface of the potential energy (Figure 2a), the minimum corresponds to a structure with the values of −60° for the torsion angles determining the position of the terminal hydroxylic groups of the glycerol residue. Then, the general position of the glycerol residue with respect to the acyl backbone is determined by varying the torsion angle $\angle C_5C_4O_2C_2$ within the step of 10°. It can be seen from Figure 2b that the structure with $\angle C_5C_4O_2C_2 = 180°$ corresponds to the minimum. Additional optimization of the 2-monoacyl-*sn*-glycerol monomer in the vicinity of the found minima allows determining more accurate values of the targeted torsion angles: $\angle C_2C_3O_3H_3 = \angle C_2C_1O_1H_1 = -61°$

and $\angle C_5C_4O_2C_2 = -176°$. These values of torsion angles enable the realization of two intramolecular hydrogen bonds, $H_1 \cdots O_2$ and $H_3 \cdots O_4$, in the found monomer structure.

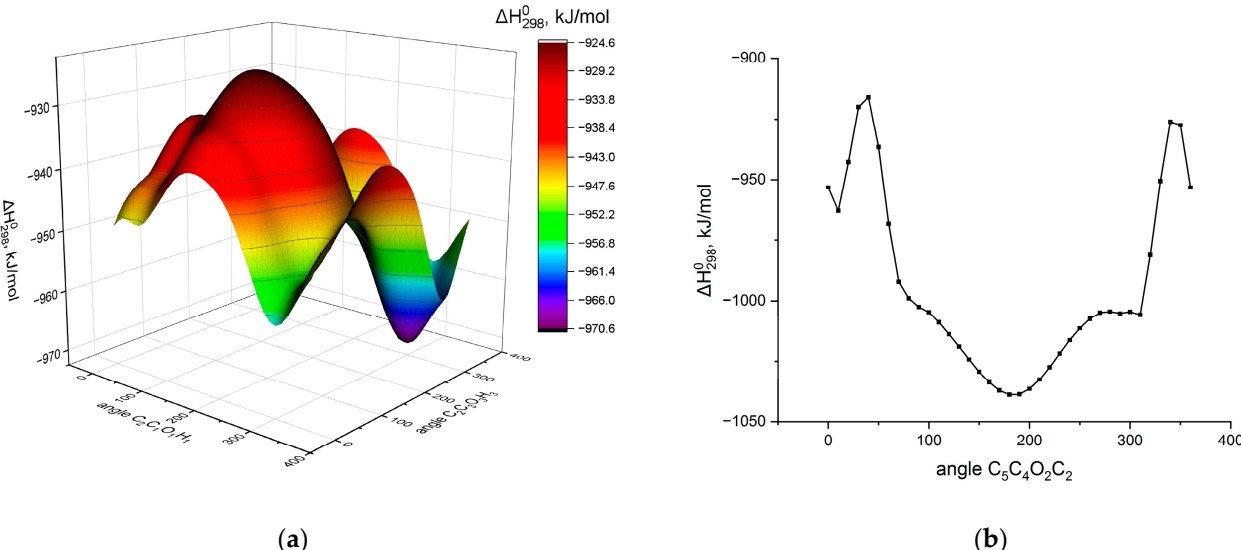

**Figure 2.** The dependence of the formation heat of 2-myristoyl-*sn*-glycerol monomer (**a**) on the values of the torsion angles of terminal hydroxylic groups; (**b**) on the value of the torsion angle of the position of the glycerol residue relative to the acyl substituent.

The conformational analysis of the 3-monoacyl-*sn*-glycerol monomer is described in detail in a previous paper [20]. Here, just its brief explanation is given. As in the case of the prior isomer, the torsion angle $\angle O_3C_3C_2C_1$ is also varied with a step of 5° in order to find the most energetically preferred orientation of the glycerol skeleton relative to the acyl substituent. The curve of potential energy for the formation of the 3-monomer was shown in [20]. It possesses two minima, $\angle O_3C_3C_2C_1 = 70°$ and $-85°$. Further additional optimization allows specifying these values as 79° and $-82°$ for Monomer 1 and 2, respectively. Both monomers also possess intramolecular hydrogen bonds between the hydrogens of the first and second carbon atoms of glycerol and carbonyl oxygen, $(C_1)H \cdots O_4$ and $(C_2)H \cdots O_4$, respectively. The third conformation of 3-monoacyl-*sn*-glycerol is also described in [20]. It is composed of glycerol conformer 1, which is less energetically advantageous. This monomer 3 is characteristic of torsion angle $\angle O_3C_3C_2C_1 = 150°$, which enables the realization of an intramolecular hydrogen bond, $H_2 \cdots O_4$.

The values of standard thermodynamic characteristics of formation are calculated for all conformations of 2 and 3-monoacyl-*sn*-glycerols [20]. Those for 2-monoacyl-*sn*-glycerols are listed in Table S1 of Supplementary Materials. The calculation showed that according to the Gibbs energy of formation, monomer 1 is the most preferred, and monomer 2 is the least preferred among the conformers of 3-monoacyl-*sn*-glycerol. Regarding the 2-monoacyl-*sn*-glycerol monomer, its formation is practically isoenergetic with the least advantageous conformation of 3-monoacyl-*sn*-glycerol. In paper [20], it is shown that monomer 1 of 3-monoacyl-*sn*-glycerol forms a more energetically preferable monolayer structure, the geometric parameters of which correspond to the available experimental data. Calculated thermodynamic data for isomers of monoacylglycerols are analyzed in order to find contributions introduced by the methylene fragment of the hydrocarbon chain and by the hydrophilic part of the surfactant molecule. As these partial correlations possess similar values of increments, it is possible to combine them into general dependence for each of the targeted thermodynamic characteristics:

$$\Delta H^0_{298,mon} = -(22.75 \pm 0.08) \cdot n - (722.88 \pm 1.14) \cdot n_{3.1} - (716.24 \pm 1.07) \cdot (n_{3.2} + n_2)$$
$$-(726.47 \pm 1.14) \cdot n_{3.3}, \text{[standard deviation S} = 1.91 \text{ kJ/mol; sample size N} = 48];$$

(1)

$$S^0_{298,mon} = (31.4 \pm 0.2) \cdot n + (384.0 \pm 2.7) \cdot (n_{3.1} + n_{3.2}) + (376.2 \pm 2.7) \cdot (n_{3.3} + n_2),$$
$$[S = 4.8 J/(mol \cdot K); N = 48]; \tag{2}$$

$$\Delta G^0_{298,mon} = (8.48 \pm 0.05) \cdot n - (594.18 \pm 0.72) \cdot (n_{3.1} + n_{3.3}) - (585.21 \pm 0.72) \cdot (n_{3.2} + n_2),$$
$$[S = 1.28 kJ/mol; N = 48], \tag{3}$$

where $n$ is the number of carbon atoms in the alkyl chain; $n_{3.i}$ is the identifier of the contribution of the hydrophilic parts of the $i$th conformer for the 3-isomer, and $n_2$ means the same for the 2-isomer of monoacyl-*sn*-glycerol molecules. The correlation coefficient is $R > 0.9999$ for mentioned regressions, and standard deviations are within the limits of those assessed for other surfactant classes.

### 3.2. Dimers, Tetramers, and Other Small Clusters

Dimers are constructed using the monomer conformations described above for both structural isomers of monoacyl glycerols. As before, dimers are singled out in two types according to the mutual orientation of the "heads". The spatial arrangement of vectors determining the monomer hydrophilic "head" orientation is the criterion for dividing 2-monoacyl-*sn*-glycerol dimers into "parallel" (p) and "sequential" types (see Figure 3). The arrow indicates a projection of the vector drawn through the centers of the carbon atoms $C_1$ and $C_3$ of the glycerol backbone. For 3-monoacyl-*sn*-glycerol, a conditional vector is carried out through the centers of oxygen atoms $O_4$ and $O_1$. The name of the dimer type for 2-monoacyl-*sn*-glycerol is given in such a way: the letters "p" or "s" are present in the name of the dimer, and the number designates the basic monomer conformation. For example, the denotation "Dimer 1, s" specifies that the dimer structure comprises Monomer 1 of 3-monoacyl-*sn*-glycerol with "sequentially" oriented hydrophilic "heads".

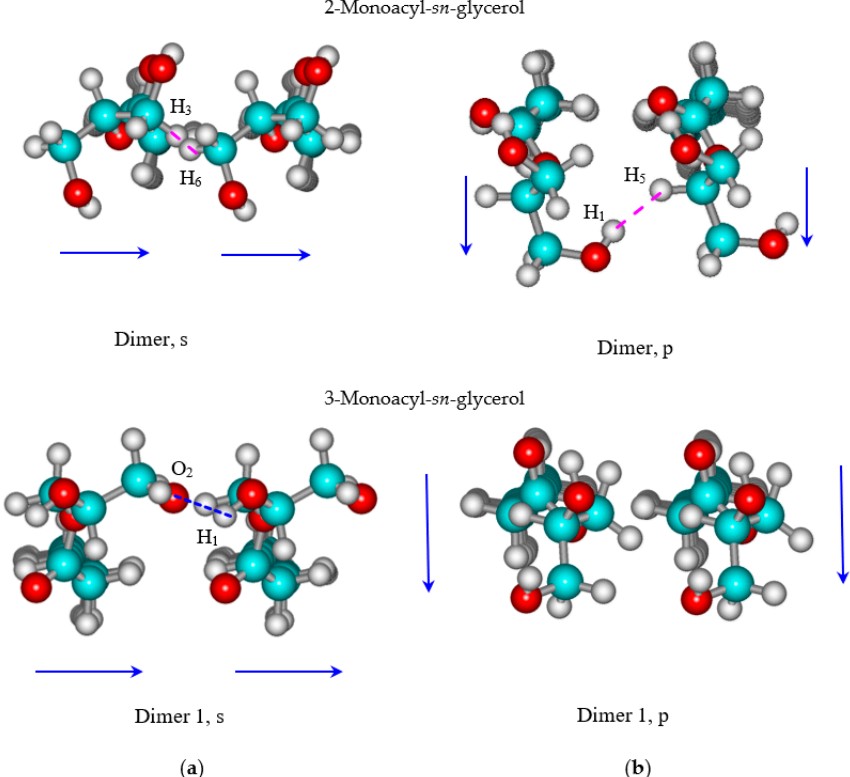

**Figure 3.** The dimer structures of monohexadecyl-*sn*-glycerol (**a**) with "sequential" and (**b**) "parallel" orientation of the hydrophilic parts.

Described types of dimers are subject to a preliminary analysis of the dimerization thermodynamic parameters in order to define if the most energetically preferred monomer conformation gives the minimal dimerization Gibbs energy. There are cases of α-amino acid [40] and α-hydroxylic acid [41] monolayers when the monomers without the most energetically advantageous conformations give the most preferred structures of the 2D films in accordance with the existing experiment. In addition, the possible geometry of dimers as basic units for larger associates (tetramers) should be specified, particularly the monomer tilt angles with respect to the normals to the axes of the monolayer propagation. This analysis was performed in Reference [20]. It showed that the most preferred Gibbs dimerization energy is typical for a pair of dimers comprised by the Monomer 1 of 3-monopentadecyl-*sn*-glycerol. In addition, it is possible to construct dimers with a different number of CH···HC interactions in them and, consequently, a different tilt angle of the monomer hydrophobic chains with respect to the normal to the axes of the monolayer propagation. As a rule, the maximum possible number of CH···HC interactions of "a" type [42,43] realized between the chains of surfactant monomers provides greater dimer preference and a smaller value of the molecular tilt angle in it [44]. At the same time, the possible number of realized CH···HC interactions depends on the geometric dimensions of the hydrophilic part of the surfactant molecule. The more voluminous the hydrophilic part of the surfactant is, the greater one monomer molecule should be displaced with respect to another in order to overcome steric hindrances in their arrangement, which governs possible loss of intermolecular CH···HC interactions.

Figure 3 presents the bottom views of the dimer structures for distinguished isomers of monoacylglycerols. The procedure for finding the optimal dimer structure is described in detail elsewhere [45]. The values of the tilt angles for Dimer, p and Dimer, s of 2-monoacyl-*sn*-glycerol are estimated as $\delta = 9.2°$ and $\varphi = 21.7°$ in the *p* and *q* directions of the monolayer propagation correspondingly (Figure 4a,b). Both dimer structures include intermolecular interactions between the hydrophilic "heads". In particular, Dimer, p contains the intermolecular interaction $O_1H_1···H_5$ between the hydrogen atom belonging to the first hydroxylic group and the hydrogen atom at the second carbon atom of the glycerol residue. Dimer, s is stabilized by the CH···HC interaction between hydrogen atoms at the first glycerol carbon atom of one monomer and the third glycerol carbon atom of the second monomer in the dimer ($H_3···H_6$). Both described interactions are marked with lilac dotted lines in Figure 3. In the case of the 3-isomer of substituted glycerol, its hydrocarbon chains are inclined with angles $\delta = 21.6°$ and $\varphi = 9.1°$ in the *p* and *q* directions of the monolayer propagation, respectively. Here, Dimer 1, s has an intermolecular hydrogen bond $H_1···O_2$ (marked with a blue dotted line in Figure 3) in accordance with the investigation by Pantoja-Romero et al. [19]. Note that the number of intermolecular CH···HC interactions between the surfactant hydrocarbon chains depending on the surfactant chain length (*n*) can be calculated as follows:

$$K_a = \left\{\frac{n-1}{2}\right\} \text{ and } K_a = \left\{\frac{n}{2}\right\} \text{ for 2-monoacyl-}sn\text{-glycerol,} \qquad (4)$$

and vice versa for 3-monoacyl-*sn*-glycerol with braces {...} denoting the integer part of the number. Therefore, one can see that the structural parameters (tilt angle and number of intermolecular CH···HC interactions) of the "parallel" dimers for 2-isomers are close to those for "sequential" dimers for 3-isomers and vice versa. But, as it will be shown below, the change in substituent position within the glycerol backbone does not much affect the structural parameters of the unit cell for studied compounds.

For this purpose, the structure of a tetramer is constructed; it comprises dimers with "parallel" and "sequential" orientation of the "heads". It serves as a unit cell of the 2D monolayer (see Figure 4c). The angle θ realized between the sides of the unit cell is determined from the optimized tetramer structures. Its value is 102° and 91° for 2 and 3-isomers, respectively. Both the values of angles δ and φ, as well as θ, enable determining the molecular tilt angle (*t*) of the surfactant chain with respect to the normal to the interface

according to the procedure described in Reference [29]. The values of the molecular tilt angle *t* are 23.0° and 23.5° for 2 and 3-monoacyl-*sn*-glycerol correspondingly in accordance with existing experimental data obtained using GIXD studies for monolayers of stereospecific and racemic 2 and 3-isomers of monoacyl glycerol [10,18,46,47]. The distances between two monomers along the *p* and *q* axes of the monolayer spread in optimized tetramer structure can be considered as the sides of the monolayer unit cell, as follows: *a* = 4.91 Å, *b* = 5.00 Å for 2-monoacyl-*sn*-glycerol, and *a* = 4.82 Å, *b* = 4.92 Å for the 3-isomer. Described values of the structural parameters indicate the oblique and orthorhombic unit cells of the monolayers for considered structural isomers. As one can see, the change in acyl substituent position does not affect the structure of formed monolayers much.

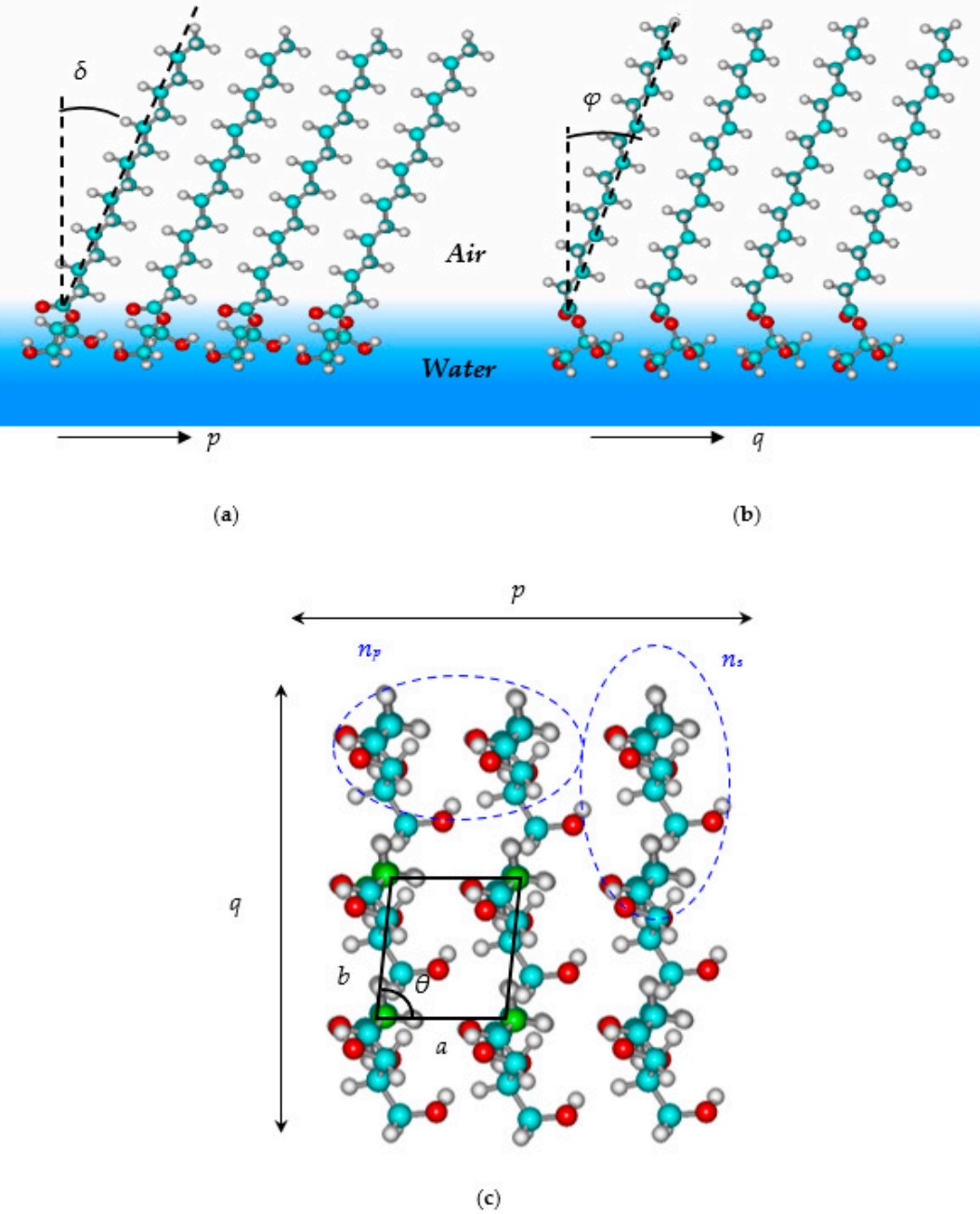

**Figure 4.** Fragment of 2D film structure for stereospecific 2-substituted monoacyl glycerol: (**a**) view along the p axis; (**b**) view along the q axis; and (**c**) view along the molecular chain axis.

Consider now if the change in the substituent position affects the thermodynamic properties of association more profoundly. The series of small clusters up to octamers are constructed and optimized for this purpose. The acyl chain length of the substituents is varied within the range of $n$ = 6–17 carbon atoms except for octamers (n = 6–10) because the calculations of the thermodynamic parameters in Mopac2000 are limited by structures with 900 atomic orbitals. Corresponding thermodynamic parameters of associate clusterization are listed in Table S2 of the Supplementary Materials for the 2-isomer. The clusterization parameters are calculated as the difference between the parameters of the formation of associates and the corresponding number of monomers. A few structures of short-chained associates are distorted for 3-monoacyl-*sn*-glycerol; therefore, their thermodynamic data are omitted in the subsequent construction of general correlation dependences. The clusterization enthalpy, entropy, and Gibbs energy depend on the number of intermolecular CH···HC interactions realized in the associate ($K_a$). Therefore, as in the cases of other surfactant classes, the partial correlations of clusterization thermodynamic parameters of linear type are constructed for the series of small clusters. The slopes of the regressions are in the range −(9.99–10.51) kJ/mol and −(16.31–25.02) J/(mol·K) for clusterization enthalpy and entropy, correspondingly. The partial correlations for small clusters for both isomers of substituted glycerols are combined into the general one below:

$$\Delta H_{298,m}^{Cl} = -(10.41 \pm 0.07) \cdot K_a - (6.56 \pm 1.33) \cdot n_p - (4.02 \pm 0.98) \cdot n_s$$
$$-(12.75 \pm 0.34) \cdot (n_{3.1,p} + n_{3.1,s}), [S = 6.08 \, kJ/mol; R = 0.9998; N = 116], \tag{5}$$

$$\Delta S_{298,m}^{Cl} = -(19.93 \pm 0.46) \cdot K_a - (150.19 \pm 2.39) \cdot (n_p + n_{3.1,p} + n_{3.1,s}) - (129.34 \pm 2.85) \cdot n_s,$$
$$[S = 42.27 \, J/(mol \cdot K); R = 0.9995; N = 116], \tag{6}$$

where $K_a$ is the number of CH···HC interactions realized in the specific cluster calculated using the equations for basic dimers (4) included in the larger clusters. Also, $n_{3.1,p}$ and $n_{3.1,s}$ are the descriptors of the "parallel" and "sequential"-oriented "head" group interactions in 3-monoacyl-*sn*-glycerol aggregate structures, whereas $n_p$ and $n_s$ are the descriptors for the same orientation types of hydrophilic groups for 2-monoacyl-*sn*-glycerols. Their values are equal to the number of interactions of this type present in a specific cluster. If such interactions are absent, then the corresponding descriptor is zero.

As one can see from Equations (5) and (6), the contribution of the interactions realized between the surfactant "heads" as in "parallel" dimers in clusterization enthalpy are larger by absolute value than for "sequential" dimers in the case of 2-substituted glycerol. However, for 3-substituted isomers, these contributions are equal, as well as in the case of clusterization entropy. In addition, mentioned contributions are larger by absolute value for the 3-isomer than for another one. As a result, according to the clusterization, the Gibbs energy aggregation of associates with "sequential head" interactions is more advantageous than "parallel" ones for 2-monoacyl-*sn*-glycerol:

$$\Delta G_{298,m}^{Cl} = -(4.47 \pm 0.21) \cdot K_a + (38.20 \pm 2.04) \cdot n_p + (34.52 \pm 1.83) \cdot n_s$$
$$+(32.04 \pm 1.05) \cdot (n_{3.1,p} + n_{3.1,s}). \tag{7}$$

However, in the case of 3-monoacyl-*sn*-glycerol, both contributions into $\Delta G_{298,m}^{Cl}$ are equal and less destabilizing than for 2-monoacyl-*sn*-glycerol. Given these values of the contributions in Equation (7), it is possible to assume that the clusterization path for 2-monoacyl-*sn*-glycerol will take place via the formation of more advantageous "sequential" dimers and their further enlargement. This can possibly provoke a more dendritic structure of monolayer formed at definite temperature with the dominance of more energetically preferred interactions of hydrophilic parts of the surfactant molecules with the further appearance of less preferable interactions. In Reference [10], the authors used Brewster angle microscopy to observe the branched fractal-like structures at the edges of 2-monoacyl-*rac*-glycerol segments. However, they are absent for the 1 and 3-monoacylglycerol domains

in spite of the fact that for both isomers, the domains are facetted, though with different shapes: 3-monoacyl-*rac*-glycerol domains are segmented with a round or cardioid shape, while 2-monoacyl-*rac*-glycerol is also segmented, but with straight lines at the edges.

### 3.3. Large Clusters and 2D Films

An additive scheme can be constructed using the contributions of CH···HC interactions between the monomer hydrophobic chains and interactions between the hydrophilic parts of surfactants to the values of clusterization thermodynamic parameters of small associates singled out in the previous paragraph. This scheme enables calculating the parameters for larger, up to infinite, surfactant clusters.

Figure 4c illustrates a fragment of the bottom view for the 2D film of 2-monoacyl-*sn*-glycerol. Here, two types of "head" group interactions realized as in "parallel" and "sequential" dimers are highlighted within the dashed lines. They are designated $n_p$ and $n_s$, respectively. Similar interactions can be distinguished for the isomer of 3-monoacyl-sn-glycerol, whose film structure comprising Monomer 1 is considered in detail in Ref. [20]. The number of interactions of the "heads" and CH···HC interactions of the surfactant "tails" can be determined using the procedure that has already become standard in the works of this series and described in detail elsewhere [31]. For both infinite 2D films of the monoacyl-*sn*-glycerol isomers (with the number of monomers $m = p \cdot q$ tending to infinity in both directions of the monolayer propagation), the dependences of the numbers of the "head" interactions ($n_{p,\infty}/m$ and $n_{s,\infty}/m$) and CH···HC interactions of the "tails" ($K_{a,\infty}/m$) per one monomer can be determined according to the formulas

$$n_{p,\infty}/m = n_{s,\infty}/m = 1 \text{ and } K_{a,\infty}/m = n - 1 \tag{8}$$

where $n$ is the number of carbon atoms in the acyl chain of the substituent.

In order to obtain expressions for the calculation of the clusterization thermodynamic characteristics per one monomer molecule in the 2D film, one should substitute the expressions (8) into the correlation equations for the calculation of the clusterization enthalpy (5) and entropy (6) as follows:

$$\Delta A_{298,\infty}^{Cl}/m = U \cdot K_{a,\infty}/m + V \tag{9}$$

where the values of the coefficients $U$ and $V$ depend on the particular thermodynamic characteristic $\Delta A_{298,\infty}^{Cl}/m$ (enthalpy, entropy, or Gibbs energy), on the structure of the monomer forming the 2D cluster and temperature. Corresponding values are summarized in Table 1.

**Table 1.** The values of the coefficients for calculation of the clusterization thermodynamic parameters per one monoacyl-*sn*-glycerol.

| Type of Infinite 2D Cluster | $\Delta H_{298,\infty}^{Cl}/m$, kJ/mol | | $\Delta S_{298,\infty}^{Cl}/m$, J/(mol·K) | | $\Delta G_{298,\infty}^{Cl}/m$, kJ/mol | |
|---|---|---|---|---|---|---|
| | $V_{\Delta H}$ | $U_{\Delta H}$ | $V_{\Delta S}$ | $U_{\Delta S}$ | $V_{\Delta G}$ | $U_{\Delta G}$ |
| 2-monoacyl-*sn*-glycerol | −10.58 | −10.41 | −279.53 | −19.92 | 72.72 | −4.47 |
| 3-monoacyl-*sn*-glycerol | −25.50 | −10.41 | −300.38 | −19.92 | 64.01 | −4.47 |

A graphical representation of the clusterization Gibbs energy for both isomers of monosubstituted glycerols on the hydrocarbon chain length at standard temperature is shown in Figure 5. Depicted dependences for thermodynamic parameters of small clusters calculated using correlations (9) are divided by the monomer number in the cluster ($m = 2$ for dimers, $m = 4$ for tetramers, etc.). This is performed in order to compare the values of the clusterization parameters for associates of different dimensions. As can be seen, spontaneous film formation of 3-monoacyl-*sn*-glycerol at standard temperature is possible for compounds with no less than $n = 16$ carbon atoms in the $C_nH_{2n+1}COO$ substituent and for 2-monoacyl-*sn*-glycerol—for molecules containing at least 18 carbon atoms in substituent.

Graphical interpretations of Equations (7) and (9) in Figure 5a,b illustrate possible ways of 2D monolayer formation for both isomers: via prevailing Dimer, s formation and their subsequent enlargement for 2-monoacyl-*sn*-glycerol, and via equiprobable formation of dimers with "parallel" and "sequential" structure for 3-monoacyl-*sn*-glycerol.

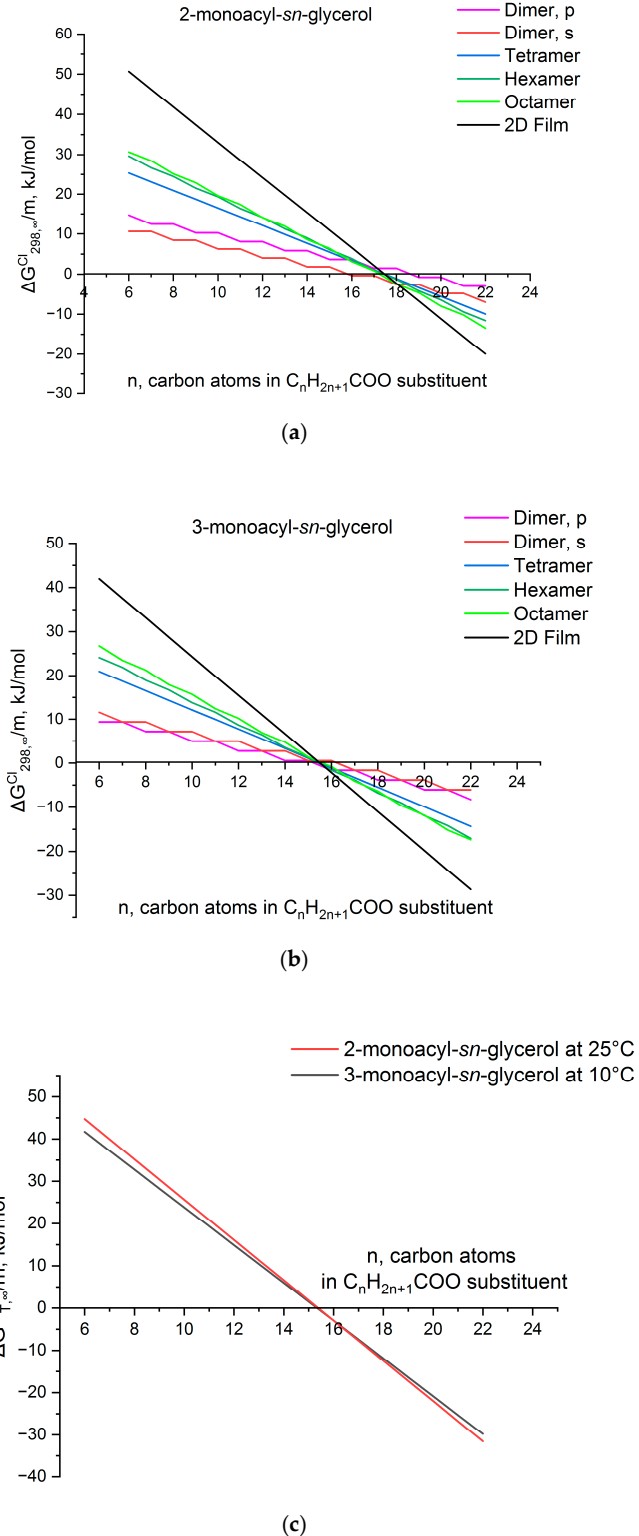

**Figure 5.** Dependence of $\Delta G_{T,\infty}^{Cl}/m$ on the chain length of monoacyl-substituted glycerols at 298 K (**a**,**b**) and on the temperature (**c**).

It is interesting to assess the temperatures enabling the monolayer formation of monoacyl-*sn*-glycerol isomers with the same chain length. This can be performed using the contributions to the clusterization enthalpy and entropy found per monomer (see Table 1). Certainly, the values of these contributions depend on the temperature. These dependencies can be taken into account within several schemes of different degrees of theoretical validity. But, as was shown in Ref. [48], even the most simplified scheme, which neglects the temperature dependences of the coefficients $V$ and $U$ for $\Delta H_{298,\infty}^{Cl}/m$ and $\Delta S_{298,\infty}^{Cl}/m$, introduces a small error in the assessment of the temperature effect of the surfactant clusterization. Therefore, the temperature is considered only as the multiplier $T$ in the next equation for clusterization Gibbs energy per monomer: $\Delta G_{298,\infty}^{Cl}/m = \Delta H_{298,\infty}^{Cl}/m - T \cdot \Delta S_{298,\infty}^{Cl}/m$. Thus, the threshold temperature of spontaneous clusterization for 2 and 3-monoacyl-*sn*-glycerol is 280 K and 279 K for compounds with $n = 16$ and 14 carbon atoms in the $C_nH_{2n+1}COO$ substituent, correspondingly. On the other hand, as can be seen in Figure 5c, 2 and 3-heptadecyl-*sn*-glycerols are capable of spontaneous film formation at 283 and 298 K, respectively, i.e., the difference between the threshold temperatures of spontaneous clusterization for these two isomers with the same chain length is $\Delta T = 15$ K, which is very consistent with the available experimental data of $\Delta T = 17.6$ K [10].

## 4. Conclusions

The thermodynamic parameters of clusterization for stereospecific monoacyl-substituted glycerol isomers with the second and third positions of the substituent relative to the glycerol residue are analyzed at the water surface. The calculations are carried out using the quantum chemical semi-empirical PM3 method. The calculations show that spontaneous film formation of these surfactants under standard conditions is possible when $n \geq 18$ and $n \geq 16$ carbon atoms in the $C_nH_{2n+1}COO$ substituent for 2 and 3-positioned surfactants, respectively. Calculated results are only 1–2 carbon atoms longer than the threshold chain length for these compounds studied using the Langmuir trough technique for obtaining $\pi$-A-isotherms. The geometric parameters of optimized tetramer structures for 2 and 3-monoacyl-*sn*-glycerol weakly differ from each other: for 2-monoacyl-*sn*-glycerol, the lengths of the sides of the oblique unit cell are slightly longer than for 3-monoacyl-*sn*-glycerol, which possesses the orthorhombic unit cell. The greater preference of contributions of "sequential"-oriented hydrophilic group interactions in clusterization Gibbs energy for 2-isomer may provoke more dendritic or fractal-like structure of condensed domains observed experimentally [10]. At the same time, the isoenergetic contribution of hydrophilic group interactions in both directions of the monolayer spread for the 3-isomer may cause an absence of any additional structures at the edges of the condensed domains. The difference between the threshold temperatures of spontaneous clusterization for these two isomers with the same chain length is $\Delta T = 15$ K, which is consistent with the available experimental data [10]. It proves the experimental fact that the change in the acyl chain substitution from the third to the second position of the glycerol backbone is almost equal to its shortening by roughly two methylene fragments.

**Supplementary Materials:** The following supporting information can be downloaded at: https://www.mdpi.com/article/10.3390/condmat8030058/s1. Table S1: Standard thermodynamic characteristics of monomer formation for 2-monoacyl-*sn*-glycerols; Table S2: Clusterization thermodynamic parameters of formation for small associates of 2-monoacyl-*sn*-glycerol in the approximation of PM3 method calculated at 298 K.

**Funding:** The study was carried out with the support of the Ministry of Science and Higher Education of the Russian Federation, the budget topic "Carbon nanoparticles with a given morphology: synthesis, structure and physico-chemical properties, FRES-2023-0006".

**Data Availability Statement:** The data that support the findings of this study are available from the author, E.K.

**Conflicts of Interest:** The author declares no conflict of interest.

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
