# Peer review of "The Effect of Acyl Chain Position on the 2D Monolayer Formation of Monoacyl-sn-Glycerol at the Air/Water Interface: Quantum Chemical Modeling"

_condensedmatter, doi:10.3390/condmat8030058_

Round 1

Reviewer 1 Report

The manuscript “Effect of acyl chain position on 2D film formation of monoacyl-sn-glycerol at the air/water interface. Quantum chemical modeling” presents a detailed study of the thermodynamics of small aggregates of glycerol derivatives. The manuscript is well presented, but I would like to present some points that, unfortunately, lead me to consider that, in the present form, the manuscript should not be publishable.

The present manuscript methodology is based on a previous contribution (by the same author - ref. 20) where the aggregation behavior of the 3-monoalkyl-sn-glycerols was explored. In that study, PM3 calculations were compared to experimental data, and a good correlation was found. In the present paper, molecules with longer chains are under study, and no experimental data supports it. I do believe that computational studies are enough in many cases to give insights into the behavior of the matter, but to do so, the method should be appropriate and validated. I do not believe that adequate methods were employed in this manuscript. To validate the findings here, I suggest molecular dynamics simulations of the systems, especially those with longer chains. The reasons are:

1.       PM3 is a semiempirical method developed based on the heat of formation to obtain the electronic properties of molecules. However, this was developed considering molecules in gas-phase or very diluted systems. So, it would not be adequate to predict the formation of films, especially in this case, where the long chains of the molecules will interact through dispersion forces. PM3 has no explicit correction for dispersion forces and should be validated (by comparing its results to experimental data or more adequate methods) for the prediction of this type of film formation;

2.       There is a slight comparison to molecular dynamics simulations performed for similar compounds (ref. 19) on page 6, lines 206-207, only for the dimer. The results for larger clusters are not comparable?

3.       I need clarification on why an extensive conformational analysis was performed for monomers. Aggregation due to hydrogen bonding and dispersion interactions can change the geometry of the monomer. Moreover, a wrong experimental value of glycerol heat of formation was used to guide the conformation selection. The author stated that “It turned out that in terms of the heat of formation, the structure of the glycerol conformer 2 (with heat of formation -605.5 kJ/mol) better corresponds to the experimental value (-607.5 kJ/mol) [23].” Ref. 23 is the Lange Handbook; the value found was -668.5 kJ/mol.

4.       Why the author assumes that chains do not change the conformational search? Is there no thermodynamic data available to compare?

5.       In the same Lange´s Handbook, one can find the surface tension for glycerol and some alkyl derivatives. Find a way to correlate the thermodynamics data of film formation predicted with PM3 to the surface tension could be a way to validate the procedure;

6.       The interaction of the polar head with water is rather important and should be included at least using continuum approaches;

7. No references were included after the following excerpts: "Calculations of regarded thermodynamic and geometrical parameters for monosubstituted glycerols are done using supermolecule approximation without explicit consideration of the interface. Its influence appears in specific conformation of the surfactant molecule done by stretching and extending action of the interface on it. This leads to predominant presence of amphiphilic moleculaes in the gas phase in the maximally elongated all-trans conformation."

[PM3]"adequately describes the threshold lengths of surfactant chains, the temperature effect of clusterization, the geometric parameters of monolayer unit cells for more than 10 classes of surfactants, as well as changes in the surface acidity or basicity of saturated carboxylic acids and amines depending on their chain length."

Author Response

I am thankful to Reviewer for his/her comments. I addressed all of them. Please see the attachment.

Reviewer 2 Report

This manuscript contains the results of theoretical studies related to the monoacyl-sn-glycerol 2D cluster formation at the air/water interface. Although the level of calculations performed might seem rather modest (PM3 semiempirical method), I believe it is justified taking into account the molecular size of the studied systems. In general, I view this manuscript as a very thorough investigation and I strongly recommend publishing this article once a few minor issues are addressed (see below):

Please elaborate on possible faults and errors in PM3 regarding the calculations performed in this contribution.

The authors wrote that “Absence of negative values of analytical harmonic vibrational frequencies is the criterion…”. Please replace “negative” with “imaginary”, as the force constants can be negative whereas the frequencies cannot (they could be imaginary as they are proportional to the square roots of force constants).

Figure 2b: it seems suspicious that the DeltaH plot increases very rapidly around the angle of 300 deg. Please comment on this.

Author Response

I am thankful to Reviewer for his/her positive decision and comments. I addressed all of them. Please see the attachment.

Reviewer 3 Report

The paper by Kartashynska is a rather minor modification of the paper Colloids and Surfaces A: Physicochemical and Engineering Aspects 667 (2023) 131400, cited as ref. [20]. It uses the same methodology as in [20], to extend the analysis to 2-monoacyl-sn-glycerols.

However, it contains large part of data about 3-monoacyl-sn-glycerols adapted from ref. [20]:
- parts of Figure 1 are the parts of Fig. 1 [20] with minor changes (color, and some lines added)
- Figure 3 is the Fig. S1 from the supplementary material of [20]
- half of Figure 4 is the Fig. 2 [20]
- Figure 5 is very similar to Fig.4 [20]
- Figure 6b is the Fig. 7 [20] with color added and the data shifted by 1 on the x-axis to reflect the different definition of the chain length
- 3 out of 4 columns of Table S1 are the exact copies of Table S1 [20]. The author carelessly forgot to copy the explanation of the values in parentheses (given in the main text of [20])
- the data in Table S3 are in fact values from Table S3 [20], but converted to a less informative form (total values instead of clusterization values)
- although the text is not an exact copy of [20], it follows it closely; the original text is more clearly written

The overlap between [20] and the submitted manuscript is so large that the latter may be regarded as self-plagiarism. Moreover, an authorship issue arises, because paper [20] has two authors and the other author is not credited in the current work. The author of the manuscript does not even bother whether "acyl" or "alkyl" compounds were studied and uses both names, which is another indication that the text might have been copied. 
Therefore the paper should be rejected. 

There are some new results in the paper regarding 2-monoacyl-sn-glycerols, leading to minor modifications of parameters in Eqs. 1-3, 5-6 and Table 1 (corresponding to Eqs. 1-3, 5-6 and 12-14 in [20]). These findings may be summarized in a short paper with proper authorship and without non-necessary repetitions of the content of the older work.

A check for style can improve the readability of the paper.

Author Response

(The authors gave the same response as above.)

Round 2

Reviewer 1 Report

The author deepened the presentation of the methods employed and the discussion of the results regarding the literature. It still needs comparison to other strategies, mainly Molecular Dynamics simulations, but the manuscript already contributes to the field.

Reviewer 2 Report

I am satisfied with the author's responces to my comments. I recommend publishing this article in its revised shape.

Reviewer 3 Report

With the changes made in revision and the explanation in the response letter I think that the paper can be accepted in the present form, although it still repeats some information which could be omitted.